# Fed-REACT: Federated Representation Learning for Heterogeneous Time Series Data

## Abstract

Motivated by high resource costs and privacy concerns that characterize centralized machine learning, federated learning (FL) emerged as an efficient alternative that allows the participating clients to collaboratively train global model while keeping their data local. In practice, distributions of clients' data vary over time and from one client to another, creating heterogeneous conditions that deteriorate performance of conventional FL algorithms. In this work, we study an FL framework where clients train on heterogeneous time series data and introduce to these settings Fed-REACT, a novel federated learning method leveraging representation learning and evolutionary clustering. The algorithm consists of two stages: (1) in the first stage, the clients learn a model that extracts meaningful features from local time series data; (2) in the second stage, the server adaptively groups clients into clusters and coordinated cluster-wise learning of task (i.e., post-representation) models for local downstream tasks, e.g., classification or regression. We provided theoretical analysis of the first stage of the proposed algorithm, and demonstrated its high accuracy and robustness in experiments on real-world time series datasets.

## 1 Introduction

Distributed training of machine learning models has helped fuel recent advances in a variety of applications including recommendation systems, image recognition, and conversational AI, to name a few. Federated Learning (FL) (McMahan et al., 2017), in particular, received significant attention as it facilitates collaborative privacy-promoting training of a global model that can subsequently be deployed on the participating clients' devices for local tasks. However, the now classical FedAvg algorithm (McMahan et al., 2017) and its variants assume independent and identically distributed (IID) data, which often does not reflect real-world scenarios. Indeed, since clients collect data locally at different times and locations, the training sets are typically heterogeneous across clients in terms of both volume and statistical distribution. Data heterogeneity has been recognized as a major challenge in federated learning (Zhao et al., 2018) – when local models are trained on non-IID data, simple (potentially weighted) averaging during aggregation generally results in underperforming global models and may lead to unacceptable performance on local tasks. Consequently, a number of techniques for mitigating the impact of data heterogeneity in FL has been explored (see, e.g., Li et al. (2020) and the references therein). Moreover, when an FL system involves a large number of clients (e.g., in cross-device scenarios), the communication overhead required to support the transmission of local updates may become prohibitive. Such large-scale settings may also be characterized by intermittent availability of the clients, rendering the coordination of the training process challenging. To this end, approaches that group clients into clusters, deploy cluster-aware sampling strategies, and ultimately train cluster-specific models, have been investigated in literature Mansour et al. (2020); Kim et al. (2021).

In many real-world applications including healthcare, autonomous driving, and finance, the data collected by clients naturally comes in the form of time series. While the above FL methods have proven effective for static heterogeneous data, most are not designed to handle time series data characterized by an additional layer of heterogeneity arising from the temporal dimension. Kim et al. (2021) proposed a framework that leverages a generative adversarial network (GAN) to group users and dynamically adjust resulting clusters without sharing raw data. However, this approach relies on clustering snapshots of temporal data, which may lead to erroneous declarations of abrupt changes to cluster membership over time. An alternative to snapshot clustering comes in the form of evolutionary

clustering (Xu et al. (2014)) which incorporates historical data to inform cluster membership decisions, generally allowing for smoother transitions and more stable clustering solutions.

## 1.1 CONTRIBUTION OF THIS WORK

The contribution of this work can be summarized as follows:

1. To the best of our knowledge, this work is the first to formally investigate the problem of federated self-supervised learning on heterogeneous time-series data. There are two sources of data heterogeneity in such FL systems: Inter-client distribution diversity, arising from the differences in data distribution across clients, and intra-client data heterogeneity, i.e., potential non-stationarity of the data observed locally by each client. We propose Fed-REACT, a novel Federated learning method leveraging Representation learning and EvolutionAry Clustering for Time-series data, that consists of two learning phases: In the first phase, which essentially deals with inter-client data heterogeneity, the clients rely on self-supervised learning to collaboratively learn meaningful features, while in the second phase, addressing intra-data heterogeneity, temporally-evolving clusters of distributionally similar clients use the extracted features to train task (i.e., post-representation) models.

2. In order to accomplish the goal of the second phase of Fed-REACT, we leverage evolutionary clustering to dynamically group clients based on the similarity of their task model weights. This is rendered difficult by the variations in those weights which are exacerbated when the training batches are small. To address this concern, we introduce an adaptive forgetting factor which facilitates clustering based on both current as well as historical weights of the task models, ensuring more accurate/stable clustering solutions. We investigate three strategies aided by adaptive forgetting: (a) time averaging; (b) weighted averaging with forgetting; and (c) Kalman smoothing utilizing expectation-maximization. The efficacy of these strategies is presented in the results section.

3. We provide theoretical analysis of feature learning on time-series data in federated learning systems. Specifically, we consider a global regret function for a linear feature model and apply time-smoothed gradient descent for time-series data. We show that with properly selected step and smoothing window size, the regret converges to a small value.

## 1.2 RELATED WORK

Federated learning allows participating clients to collaboratively train a global model while keeping the training data local and private; the clients may subsequently deploy the resulting model to local inference tasks. However, the heterogeneity of data that is generally collected at different locations and times poses significant challenges. In particular, data heterogeneity often leads to performance degradation of the trained models, motivating various efforts to address this issue.

On another note, self-supervised learning (SSL) has shown promise in distributed learning systems, particularly when handling large imbalanced datasets (Wang et al., 2022). Unlike supervised learning, SSL uses a two-stage approach: extracting features from unlabeled data, followed by utilizing these features when training for downstream tasks. While SSL has proven effective for static data in fields such as natural language processing and video processing, its applications to time series data have received less attention Chen et al. (2020); Chen & He (2021); Chen et al. (2024).

In another development, Fortuin et al. (2018) and Franceschi et al. (2019) introduced methods for learning temporal representations, for which the latter leveraged causal dilated convolution and time-based negative sampling. Wu et al. (2022) considered multi-periodicity in time series and proposed TimesNet to learn intraperiod- and interperiod-variations from temporal sequences. Nie et al. (2022) designed a Transformer-based self-supervised method, PatchTST, to improve the long-term forecasting accuracy. More recent work by Fraikin et al. (2023) and Eldele et al. (2024) has explored self-supervised approaches to capturing temporal embeddings and long- and short-term dependencies. TimeLLM Jin et al. (2023) further reprogrammed time series input into text prototype representations to adapt large language models to time series forecasting. Despite these advancements, most research on time series representation learning remains focused on centralized settings, with relatively few studies addressing distributed learning systems.

When clustering time series data, evolutionary methods aim to account for the dynamic nature of the objects being clustered. These methods often outperform snapshot clustering which only considers data at specific time points. Examples of evolutionary clustering methods include Xu et al. (2014) which introduced the Adaptive Evolutionary Clustering Algorithm (AFFECT), an iterative technique that updates a weighted affinity function to maintain temporal continuity in clustering. Arzeno & Vikalo (2019) subsequently proposed Evolutionary Affinity Propagation (EAP), a method that groups data by message exchange on a factor graph. However, EAP is limited to offline scenarios and struggles to handle streaming data effectively. In the federated learning system, clustering techniques have been employed to group clients with similar data distributions. Ghosh et al. (2020) proposed Ierative Federated Clustering Algorithm (IFCA), which determines cluster membership based on similarity coefficients. Li et al. (2021a) proposed the Federated Soft Clustering (FLSC) method, demonstrating that allowing overlapping cluster memberships can significantly enhance performance. More recently, Mehta & Shao (2023) presented an agglomerative clustering method for federated learning, which greedily identifies cluster centers through gradient updates.

### 1.3 PROBLEM STATEMENT

We consider a federated learning system with $n$ clients in which each client collects local time series data with features $x \in \mathcal{R}^{d \times T}$ and label $y$, where $d$ denotes the feature dimension and $T$ denotes the maximum length of the time series data. A server coordinates collaborative training of a global model by collecting local updates from the clients, aggregating them, and distributing the aggregated updates among the clients. The dataset at client $i$, containing the local time series data, is denoted by $\mathcal{D}_i(x, y)$. The distribution of $\mathcal{D}_i$ varies from one client to another, naturally leading to the data heterogeneity in the system. In a self-supervised learning framework, a feature-extraction function $f_\theta(\cdot)$, parameterized by $\theta$, is learned to extract the meaningful representations from the input data; this is an encoder that learns the mapping $\mathcal{R}^{d \times T} \rightarrow \mathcal{R}^{\hat{d}}$. The representations can then be utilized for downstream supervised learning tasks. Depending on the task (e.g., regression or classification), a lightweight task function $f_{\theta_{task}}(\cdot)$, parameterized by $\theta_{task}$, can be trained on the features extracted from a much smaller set of labeled samples.

The remainder of the paper is organized as follows. Section 2 presents details of the proposed method. Section 3 provides theoretical analysis of the algorithm's performance. Section 4 reports experimental results, while Section 5 concludes the paper.

## 2 ALGORITHM DEVELOPMENT

Our proposed approach is organized in two phases: in the first phase, the method learns lower level representations using a feature model while in the second phase it captures higher level features and facilitates downstream tasks. The main reasoning for such an organization is in meaningfulness of sharing the lower level feature representations of input vectors across clients regardless of their local data distributions. In the case of images, for example, two clients may own data coming from vastly different distributions; however, objects in images typically share low level features such as edges and corners. It would thus be desirable if feature model learning could include all clients regardless of local data distribution – this is enabled by training the encoder in a federated manner. Specifically, the encoder training is focused on minimizing the contrastive loss (Chen et al. (2020); Franceschi et al. (2019)). Let the reference anchor $x^{ref}$ be any given time series data, let $\{x^{neg}\}_{r=1}^R$ denote a set of $R$ randomly selected negative samples, and let $x^{pos}$ be a positive sample. Then the contrastive loss function is defined as

$$L(x^{ref}, x^{pos}, \{x^{neg}\}_{r=1}^R; \theta) = -\log(\sigma(f(x^{ref}; \theta)^T f(x^{pos}; \theta)))$$

$$- \sum_{r=1}^R \log(\sigma(-f(x^{ref}; \theta)^T f(x^{neg(r)}; \theta))), \quad (1)$$

where $f(\cdot; \theta)$ denotes the output of the encoder parameterized by $\theta$ and $\sigma(\cdot)$ denotes the sigmoid function. Minimization of the loss function ensures that the features extracted from the anchor $x^{ref}$ and its positive sample are similar to each other, while the features extracted from the anchor and its negative samples differ from each other. For time series data, the positive sample is a sub-sequence

from the same trajectory, while the negative samples are sub-sequences from other trajectories. The encoder being used is a Causal CNN with exponentially dilated convolutions, known to capture long range dependencies more effectively than full convolutions (Franceschi et al., 2019). The complete federated representation learning procedure is formalized as Algorithm 1.

In the second phase, the focus is shifted to downstream tasks. The task model captures higher level features specific to local data properties; it is therefore meaningful that clusters of clients with similar data distributions collaboratively learn shared task model weights. The choice of the architecture of a task model is driven by the downstream task category: for classification tasks we adopt SVMs, while for regression problems a simple linear layer trained using an $\ell_2$ loss function can be deployed. Note, however, that clients cannot communicate label distributions due to privacy concerns; as an alternative, we pursue clustering of the clients based on the weights of their respective task models. A simple approach could be that the server collects task model weights from clients in each round of training and employs Agglomerative Hierarchical Clustering to organize the clients into clusters. Detailed description of this approach to clustering is formalized as Algorithm 3 in the appendix.

This clustering method, however, only considers snapshot of temporal data and is incapable of accounting for the correlations within time series. Further challenges stem from the following:

1. The number of labeled samples used to train a task model is much smaller than the number of unlabelled samples used to train the encoder.

2. Typically, clients can store labelled data only for a limited amount of time before the data is deleted or replaced by newly collected samples.

Consequently, the task models trained in a single round (i.e., on a temporal snapshot of time series data) may not be sufficiently reflective of the local data distributions, ultimately also leading to incorrect clustering results. To make the clustering phase of our algorithm robust to training variations, we rely on Adaptive Evolutionary Clustering (Xu et al., 2014) where the clusters are allowed to evolve over time. Let us define the underlying similarity matrix at time $t$, $\psi_t$, which captures client relationships within and across clusters. The observed similarity matrix, $W_t$, is a noisy version of $\psi_t$, i.e.,

$$W_t = \psi_t + N_t, \tag{2}$$

where each element of $W_t$, $[W_t]_{i,j}$, denotes the cosine similarity between the vectorized parameters of task models of clients $i$ and $j$, and where $N_t$ denotes the noise. Evolutionary Clustering Algorithm (Chakrabarti et al. (2006)) incorporates the estimate of the similarity matrix at time $t-1$, $\hat{\psi}_{t-1}$, using a forgetting factor $\alpha$, to obtain the current estimate

$$\hat{\psi}_t = \alpha\hat{\psi}_{t-1} + (1-\alpha)W_t, \tag{3}$$

with initial $W_0 = 0$. Adaptive Evolutionary Clustering Algorithm (AFFECT) by Xu et al. (2014) builds upon this to propose an algorithm that iteratively estimates the forgetting factor at each time instant to obtain both $\alpha_t$ and $\hat{\psi}_t$,

$$\hat{\psi}_t = \alpha_t\hat{\psi}_{t-1} + (1-\alpha_t)W_t. \tag{4}$$

Once the estimate $\hat{\psi}_t$ is obtained, one can assign cluster membership to the clients using Agglomerative Heirarchical Clustering as described previously.

---

**Algorithm 1** Fed-REACT Phase 1: Encoder training

---

1: **Input:** Number of rounds $T$, number of clients $K$, initialized global encoder parameters $\theta_0$
2: **for** each round $t = 1, 2, ..., T$ **do**
3:     **for** each client $k = 1, 2, ..., K$ **do**
4:         Client $k$ downloads current global model parameters $\theta_{t-1}$
5:         Client $k$ updates parameters $\theta_t^k$ using local time series data
6:         Client $k$ uploads updated parameters $\theta_t^k$ to the server
7:     **end for**
8:     Server aggregates collected updates as

$$\theta_t = \sum_{k=1}^{K} \frac{n_k}{n}\theta_t^k,$$

    where $n_k$ is the number of samples on client $k$ and $n = \sum_{k=1}^{K} n_k$
9: **end for**

---

Having discovered temporal dynamics of the underlying clusters of clients allows us to explore several strategies for combining weights of task models (e.g., SVM parameters) calculated in different rounds of training. In particular, we explore the following three approaches to combining parameters of the cluster-specific task models evaluated throughout the training process:

1. **Approach 1: Simple Temporal Averaging (A1).** Parameters of the task model are obtained by taking the sample mean, i.e.,

$$\hat{\theta}_{task,t+1}^c = \frac{t}{t+1}\hat{\theta}_{task,t}^c + \frac{1}{t+1}\theta_{task,t}^c.$$

(5)

Here, $\theta_{task,t}^c$ denotes the parameters of the task model for cluster $c$ computed solely based on the temporal snapshot of data at time $t$, while $\hat{\theta}_{task,t}^c$ denotes the parameters computed based on $\theta_{task,1}^c, \theta_{task,2}^c, ..., \theta_{task,t}^c$. The initial value $\hat{\theta}_{task,1}^c$ is set to $\theta_{task,1}^c$.

2. **Approach 2: Weighted Averaging with Forgetting (A2).** In this approach we use the adaptive forgetting factor $\alpha_t$ returned by the evolutionary clustering algorithm to update the weight estimate according to

$$\hat{\theta}_{task,t+1}^c = \alpha_t\hat{\theta}_{task,t}^c + (1-\alpha_t)\theta_{task,t}^c$$

(6)

3. **Approach 3: Kalman Smoothing with Expectation Maximization (A3).** In this approach, we treat clustering solutions up to time $t$, $\{\theta_{task,s}^c\}_{s=1}^t$, as "measurements", and find the optimal linear estimate of $\theta_{task,t}^c$ via the Kalman Filter (Welch et al., 1995). In other words, we think of $\{\theta_{task,s}^c\}_{s=1}^t$ as if they were noisy observations of the true parameters of the task model, evolving according a state space model with an unknown state transition matrix $F$, innovation noise covariance $Q$, and measurement noise covariance $R$. These unknown parameters are iteratively estimated via the Expectation-Maximization algorithm (Shumway & Stoffer, 1982); details are provided in the appendix.

---

**Algorithm 2** Fed-REACT Phase 2: Task model training with evolutionary clustering

---

1: **Input:** Number of rounds $T_{task}$, number of clients $K$, cluster number $C$, trained encoder $\theta_T$
2: **for** each round $t = 1, 2, ..., T_{task}$ **do**
3:    **for** client $k = 1, 2, .., K$ **do**
4:       Client $k$ trains the task model on randomly sampled local dataset $\mathcal{M}_t^k$
5:       Client $k$ uploads the parameters $\theta_{task,t}^k$ of the task model to the server
6:    **end for**
7:    Server clusters clients based on the weights of the task models $\{\theta_{task,t}^k\}_{k=1}^K$ using AFFECT algorithm to obtain the cluster membership of $C$ clusters, $\{\mathcal{S}_t^c\}_{c=1}^C$ and adaptive forgetting factor $\alpha_t$.
8:    **for** cluster $c = 1, 2, .., C$ **do**
9:       Server aggregates the task models of all clients within cluster $\mathcal{S}_t^c$

$$\theta_{\mathbf{task,t}}^{\mathbf{c}} = \sum_{k\in\mathcal{S}_t^c} \frac{|\mathcal{M}_t^k|}{\sum_{j\in\mathcal{S}_t^c}|\mathcal{M}_t^j|}\theta_{task,t}^k$$

10:       **if** $t \geq T_{task}$ or $\mathcal{S}_t^c = \mathcal{S}_{t-1}^c$ **then**
11:          Compute $\hat{\theta}_{\mathbf{task,t}}^c$ using Approach A1, A2 or A3
12:          Server transmits $\hat{\theta}_{\mathbf{task,t}}^c$ to all clients $k \in \mathcal{S}_t^c$
13:       **end if**
14:    **end for**
15: **end for**

---

## 3 THEORETICAL ANALYSIS

In this section, we provide theoretical insights for the first phase of Fed-REACT algorithm, i.e., representation learning to heterogeneous time-series data. In particular, we focus on the convergence property of the time-varying objective function under assumption that each client trains a linear encoder via the dynamic time-smoothed gradient method. For the sake of tractability, we consider

the SSL formulation simplified from equation 1 and utilizing a local loss function defined as

$$f_{SSL,k}(\theta) = -\mathbb{E}[(\theta(x_{k,i}) + \xi_{k,i})^T(\theta(x_{k,i}) + \xi'_{k,i})] + \frac{1}{2}\|\theta^T\theta\|^2$$

at client $k$, where $\xi_{k,i}$ and $\xi'_{k,i}$ denote random noise added to the data sample $x_{k,i}$, while the global objective is defined as

$$f_{SSL} = \sum_{k=1}^{K} \frac{|\mathcal{D}_k|}{|\mathcal{D}|} f_{SSL,k}(\theta).$$

This objective is a variant of the contrastive loss equation 1 obtained by replacing the normalization via negative signals by an alternative regularization term. Optimizing $f_{SSL}$ is equivalent to minimizing $f(\theta) = \|\bar{X} - \theta^T\theta\|^2$, where $\bar{X} = \sum_k \frac{|\mathcal{D}_k|}{|\mathcal{D}|} X_k$ and $X_k = \mathbb{E}_{x \sim \mathcal{D}_k}(xx^T) = \frac{1}{|\mathcal{D}_k|}\sum_{i=1}^{|\mathcal{D}_k|} x_{k,i}x_{k,i}^T$, the empirical covariance matrix of client $k$'s data (Wang et al., 2022).

To proceed with the analysis, we make the following assumptions regarding the local loss function.

**Assumption 1.**      *1. Each loss function $f_{t,i}$ is bounded above by $M$ for all clients $i$ and times $t$.*

     *2. Each loss function $f_{t,i}$ is L-Lipschitz and $\beta$-smooth.*

     *3. Each stochastic gradient descent $\tilde{\nabla}f(\cdot)$ is unbiased and the standard deviation of the estimator is bounded above by $\sigma^2$. The error between the projected stochastic gradient $Proj\tilde{\nabla}f(\cdot)$ and the stochastic gradient $\tilde{\nabla}f(\cdot)$ is $\epsilon_{proj} = Proj\tilde{\nabla}f(\cdot) - \tilde{\nabla}f(\cdot)$ with $\|\epsilon_{proj}\|^2 \leq \epsilon^2$.*

Jin et al. (2017) have shown that the form of the objective function studied in our work is $16\Gamma$-smooth within the region $\{x | \|x\|^2 \leq \Gamma\}$ for $\Gamma \geq \lambda_1(\bar{X})$, implying that the first two assumptions are readily satisfied. Note that the projected gradient applied by the proposed algorithm guarantees that $x$ remains within the region at all time steps. The last assumption is standard in optimization literature.

Next, let us specify the update rule applied by client $k$ during the encoder learning phase. Specifically, the updates follow time-smoothed gradient descent (Aydore et al., 2019), i.e., the local update is

$$\theta_{t+1,k} = \theta_t - \frac{\eta}{W}\sum_{j=0}^{w-1} \gamma^j Proj\tilde{\nabla}f_{t-j,k}(\theta_{t-j})$$

while the global update is found as

$$\theta_{t+1} = \frac{1}{n}\sum_{k=1}^{K} \theta_{t+1,k},$$

where $w$ denotes the smoothing window size, $W = \sum_{j=0}^{w-1} \gamma^j$ and $\eta$ is the step size. Moreover, we define the local regret at client $k$ and the global regret as

$$S_{t,w,\gamma,k}(\theta_t) = \frac{1}{W}\sum_{j=0}^{w-1} \gamma^j f_{t-j,k}(\theta_{t-j})$$

and

$$S_{t,w,\gamma}(\theta_t) = \frac{1}{K}\sum_{k=1}^{K} \frac{1}{W}\sum_{j=0}^{w-1} \gamma^j f_{t-j,k}(\theta_{t-j}),$$

respectively. It holds that

$$\mathbb{E}[\tilde{\nabla}S_{t,w,\gamma}(\theta_t)|\theta_t] = \nabla S_{t,w,\gamma}(\theta_t), \quad \mathbb{E}[\tilde{\nabla}S_{t,w,\gamma,k}(\theta_t)|\theta_t] = \nabla S_{t,w,\gamma,k}(\theta_t),$$

and that

$$\mathbb{E}[\tilde{\nabla}S_{t,w,\gamma,k}(\theta_t) - \nabla S_{t,w,\gamma,k}(\theta_t)|\theta_t] \leq \frac{\sigma^2(1-\gamma^{2w})}{W^2(1-\gamma^2)}.$$

With this notation in place, we can obtain the following Lemmas and Theorem 1.

**Lemma 1.** *Suppose all of the above assumptions are satisfied. Then for any $\gamma \in (0,1)$, $\beta$ and $\eta$, it holds that*

$$(\frac{\eta}{4} - \frac{\eta^2\beta}{8})\|\nabla S_{t,w,\gamma}(\theta_t)\|^2 \leq S_{t,w,\gamma}(\theta_t) - S_{t+1,w,\gamma}(\theta_{t+1}) + S_{t+1,w,\gamma}(\theta_{t+1}) - S_{t,w,\gamma}(\theta_{t+1})$$

$$+ \eta^2\frac{\beta}{4}\frac{\sigma^2(1-\gamma^{2w})}{W^2(1-\gamma^2)} + (\frac{\eta}{4} - \frac{\eta^2\beta}{8} + \frac{\eta^2\beta}{2})\epsilon^2.$$

**Lemma 2.** *Suppose all of the above assumptions are satisfied. Then for any $\gamma \in (0,1)$ and $w$, it holds that*

$$S_{t+1,w,\gamma}(\theta_{t+1}) - S_{t,w,\gamma}(\theta_{t+1}) \leq \frac{M(1+\gamma^{w-1})}{W} + \frac{M(1-\gamma^{w-1})(1+\gamma)}{W(1-\gamma)}.$$

**Lemma 3.** *Suppose all of the above assumptions are satisfied. Then for any $\gamma \in (0,1)$ and $w$, it holds that*

$$S_{t,w,\gamma}(\theta_t) - S_{t+1,w,\gamma}(\theta_{t+1}) \leq \frac{2M(1-\gamma^w)}{W(1-\gamma)}.$$

**Theorem 1.** *Suppose all of the above assumptions are satisfied. When $\eta = \frac{1}{\beta}$, $\gamma \to 1^-$, it holds that*

$$\lim_{\gamma \to 1^-} \frac{1}{T}\sum_{t=1}^{T}\|\nabla S_{t,w,\gamma}(\theta_t)\|^2 \leq \frac{1}{W}(64\beta M + 2\sigma^2) + \frac{5}{8}\epsilon^2.$$

The theorem implies that when an appropriate step size and window size $w$ are selected, the upper bound is dominated by the second term, i.e., the projection error between the stochastic gradient and the projected gradient. Therefore, the global regret approaches a (small) value specified by the gradient projection error.

## 4 EXPERIMENTS

### 4.1 EXPERIMENTS ON THE RTD DATASET

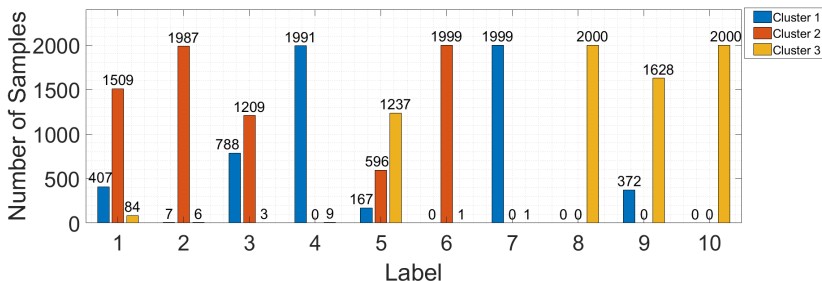

Figure 1: Label distribution for the three clusters generated using $\beta = 0.1$.

We first evaluate our proposed scheme on the RTD dataset (Alam et al., 2020) which contains 3D air-writing trajectories for 2000 samples of each digit $(0-9)$. The trajectories vary in length, with a maximum length of 100; shorter sequences are zero-padded to reach the maximum length.

The dataset is partitioned into three clusters, generated using Dirichlet distribution with a parameter $\beta = 0.1$ which leads to highly heterogeneous clusters. An example of label distribution is shown in Fig. 1; there, Cluster 1 is primarily composed of digits 3 and 6, Cluster 2 contains digits 0, 1, 2, and 5, while Cluster 3 consists of digits 4, 7, 8, and 9.

Table 1: A comparison of self-supervised and supervised learning.

| Number of clients | 10 | 50 |
|---|---|---|
| LSTM - FedAvg | 0.732 | 0.945 |
| LSTM - Fedprox | 0.804 | 0.896 |
| LSTM - Ditto | 0.863 | 0.859 |
| LSTM - APFL | 0.828 | 0.946 |
| Algorithm 1 + SVM | **0.992** | **0.948** |

### 4.1.1 SELF-SUPERVISED VS. SUPERVISED MODELS

The first set of experiments compares the performance of self-supervised and supervised baselines trained and tested on heterogeneous time series data. When the system has $K = 10$ clients, Cluster 1 and Cluster 2 each contain three clients while Cluster 3 contains four; when $K = 50$, Clusters 1, 2, and 3 comprise of 16, 16, and 18 clients, respectively. The local datasets are further divided into training and testing sets, with a $90/10$ split. As benchmarking algorithms we use a supervised learning model – single-layer LSTM model with a feature embedding dimension of $128$ and a hidden size of $256$. Each client performs local supervised training for $100$ epochs with a batch size of $50$, using the Adam optimizer with a learning rate of $0.001$. A total of $10$ communication rounds are conducted, with model aggregation performed at the server. We investigated the following state-of-the-art methods designed for federated learning with data heterogeneity.

- Fedprox by Li et al. (2020).
- Ditto by Li et al. (2021b). We set the regularization parameter $\lambda$ to $0.0001$ and the number of accumulation steps to $10$.
- Adaptive Personalized Federated Learning (APFL) by Deng et al. (2020). Here we set $\alpha$ and $\alpha_{adaptive}$ to $0.5$ and $1$, respectively.

For the self-supervised learning model trained via Algorithm 1 we consider causal time dilated CNNs; this encoder consists of ten 1D convolutional blocks, with dilation increasing by a factor of two with each layer. Each block uses leaky ReLU activation (negative slope $0.01$), followed by a linear layer that outputs features of size $320$. The encoder is trained using contrastive loss as outlined in (Franceschi et al., 2019). The task model is a SVM classifier that predicts one out of ten classes based on the encoded features. Each client performs $500$ training steps per communication round, with a batch size of $10$, using the Adam optimizer with learning rate $0.001$.

The results in the second column of Table 1 demonstrate that in the system with $10$ clients, the self-supervised model significantly outperforms supervised baseline methods in the considered data-heterogeneous scenario. The third column shows that our proposed approach maintains the superior performance over baseline methods in the larger systems that involves $50$ clients.

### 4.1.2 CLUSTERING PERFORMANCE

The second set of experiments evaluates different clustering methods and validates the performance of Algorithm 2. The Dirichlet distribution parameter is set to $\beta = 1.5$. The training of task models uses $|\mathcal{M}_t^k| = 64$ labeled samples for client $k$ at time $t$; a total of $60$ communication rounds are conducted. The baseline clustering methods include snapshot clustering (i.e., clustering based on the current values of the task model coefficients) and IFCA (Ghosh et al. (2020)) where the cluster membership is based on the similarity coefficients. Quality of a clustering solution is characterized by the Rand Score between the cluster memberships obtained from the weights of the task model and the ground truth. Recall that the Rand Score is computed as follows: Let TP be the number of pairs of clients correctly placed in the same cluster by an algorithm, let TN be the number of pairs of clients correctly placed in different clusters, and let TOT denote the total number of possible pairs of clients; then the Rand Score is calculated as (TP+TN)/TOT. Fig. 2 and Fig. 3 show the results for 10 and 100 clients, respectively; for the latter, Clusters 1, 2 and 3 contain 33, 33 and 34 clients, respectively. Fig. 2 demonstrates that Fed-REACT converges to the ground truth in as few as 3 communication rounds, while snapshot clustering method struggles to

Table 2: Clustering performance in terms of accuracy (averaged across clients). SC stands for snapshot clustering, EC stands for evolutionary clustering.

| Number of clients | 10 | 100 |
|---|---|---|
| SC (No Past Value) | 0.763 | 0.716 |
| EC (No Past Value) | 0.859 | 0.737 |
| Fed-REACT w/ A1 | 0.909 | 0.750 |
| Fed-REACT w/ A2 | 0.928 | **0.751** |
| Fed-REACT w/ A3 | **0.943** | 0.739 |
| IFCA | 0.774 | 0.740 |
| FLSC | 0.83 | 0.729 |

discover the ground truth due to training variations. The Rand Score of IFCA is a constant $0.2667$ and is omitted from the figure. When the number of clients increases to 100, the Rand Score of Fed-REACT still converges to the ground truth while the baselines suffer from oscillations and fail to approach the ground truth.

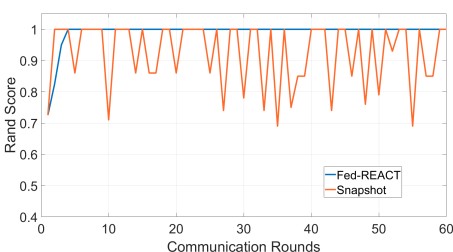 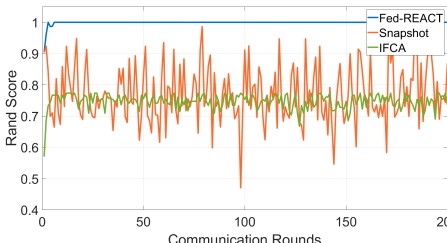

Figure 2: Rand Index Score vs. the ground truth for Fed-REACT (our method) and the baseline clustering methods (the system has 10 clients).

Figure 3: Rand Index Score vs. the ground truth for Fed-REACT (our method) and the baseline clustering methods (the system has 100 clients).

The next set of experiments, obtained on the RTD dataset, compares the accuracy of the clustering assignments for the aforementioned settings with 10 and 100 clients. Specifically, for each algorithm we calculate the instantaneous accuracy averaged over 60 rounds. Apart from snapshot clustering and the IFCA method, we also include among baselines FL with Soft Clustering (FLSC) Li et al. (2021a).[1] For Algorithm 2, we compare the accuracy obtained using the three approaches to computing the weights of SVM discussed above. For Approach A3, we set $R = rI$, $Q = qI$, $F = I$, and $P_0 = I$, and perform a grid search over $r, q \in [0.001, 0.01, 0.1, 1, 10]$. For completeness, we also include the results obtained while ignoring past values of the task model weights.

For the above two baselines, we perform simple averaging across rounds (Approach A1). The results are presented in Table 2. The second column, reporting results for the system with 10 clients, indicates that by including historical information, evolutionary clustering methods are capable of discovering the true structure of the clusters and generally achieve higher accuracy than snapshot clustering techniques. Approach A3 further improves the performance of Fed-REACT in this system. The last column in Table 2 considers an FL system with 100 clients. The representation learning phase (Algorithm 1) is carried out for 10 rounds, while the task model (i.e., SVM) is trained for 200 rounds. Note that since the grid search over the initialization for 100 clients proved to be expensive, we reused the initialization for Approach A3 obtained for the experiments involving 10 clients. This may in part explain why in this setting the performance of Fed-REACT using Approach A3 lags behind that of Fed-REACT using one of the first two approaches.

### 4.1.3 ABLATION STUDY

Lastly, we perform an ablation study exploring the relationship between heterogeneity, controlled by the parameter $\beta$, and the achieved accuracy averaged across clients. To reiterate, smaller value of $\beta$ induces greater level of heterogeneity across clusters. We consider the FL system with 100 clients; the number of clients per cluster remains the same as in the previous experiments. Since in the setting with 100 clients Approach A3 lagged in performance behind Approaches 1 and 2, we exclude the former from the ablation study. The results are presented in Table 3. As can be seen there, benefits of clustering are more pronounced for highly heterogeneous settings. As the heterogeneity across the clusters decreases, benefits of clustering diminish and the performance deteriorates.

| $\beta$ | SC (No Past Value) | EC (No Past Value) | Fed-REACT w/ A1 | Fed-REACT w/ A2 | IFCA | FLSC |
|---|---|---|---|---|---|---|
| 0.10 | 0.887 | 0.887 | 0.888 | **0.900** | 0.889 | 0.693 |
| 0.25 | 0.868 | 0.868 | **0.872** | 0.871 | 0.872 | 0.761 |
| 0.50 | 0.809 | 0.809 | **0.816** | 0.815 | 0.711 | 0.735 |
| 2.0 | 0.712 | 0.721 | **0.742** | 0.738 | 0.730 | 0.721 |

Table 3: The effect of heterogeneity on the performance. SC stands for snapshot clustering while EC stands for evolutionary clustering.

---

[1]The Rand Score for FLSC could not be calculated as each client is assigned to more than one cluster.

## 4.2 EXPERIMENTS ON THE SUMO EV DATASET

In this section, we consider Simulation of Urban Mobility (SUMO) dataset (Krajzewicz et al., 2012). This set consists of data emulating vehicles driving under varying conditions including temperature, humidity, elevation, and location. The task, unlike in the previous experiments, is at core a regression – in particular, the goal is to predict the percentage of battery life available given the 100-step multivariate time series data as the input. Consequently, while the encoder architecture remains the same as before, instead of SVM we use a linear output layer. The vehicles in the dataset have vastly different data amounts, ranging from just above 100 for some to more than 1000 training samples for others. The battery life differs even among vehicles of the same type, presenting further challenge to the client clustering task. The time series data include information about latitude, longitude, elevation, temperature, speed, maximum possible speed, acceleration, and vehicle type. The features are normalized before being fed into the models. The dataset is divided into the training and testing subsets, with a 90/10 split; there are 50 vehicles in the test set. The number of clusters is varied from $C = 1$, indicating no personalization, to $C = 50$, corresponding to the complete personalization of the output layer.

Similar to the experiments involving the RTD dataset, we compare Fed-REACT with the LSTM baselines. A crucial difference, however, is that for SUMO dataset we do not a priori know the number of clusters. This is why we test the performance of our method for various values of C, the total number of clusters, with $C = 1$ denoting global averaging of the output layer and $C = 50$ denoting complete personalization. The root mean-square error (RMSE) averaged across clients is presented in Table 4. As can be seen from the table, the higher the level of personalization, the lower the incurred RMSE. These results suggest that while federated learning of representation models on SUMO dataset greatly helps extract meaningful features from the temporal data therein, the time series generated by different vehicles is exceedingly heterogeneous thus warranting fully personalized output layers.

Table 4: Performance on SUMO EV dataset: Fed-REACT vs. supervised learning baselines.

|  | RMSE |
| --- | --- |
| LSTM - FedAvg | 43.2 |
| LSTM - Fedprox | 42.3 |
| LSTM - Ditto | 42.0 |
| LSTM - APFL | 42.7 |
| Fed-REACT (C=1) | 24.4 |
| Fed-REACT (C=3) | 23.7 |
| Fed-REACT (C=9) | 13.0 |
| Fed-REACT (C=25) | 8.8 |
| Fed-REACT (C=40) | 5.8 |
| Fed-REACT (C=50) | **1.3** |

## 5 CONCLUSION

In this paper, we studied the problem of federated self-supervised representation learning complemented by (semi)personalized task model training. This is, to our knowledge, the first work to consider such a learning problem in the setting where clients' data are heterogeneous time series. The proposed scheme, Fed-REACT, aggregates representation models globally and performs cluster-wise aggregation of task models (e.g., SVMs for classification tasks and dense output layers for regression). Convergence of the proposed representation learning scheme was studied theoretically, while experimental results on RTD and SUMO EV datasets demonstrated advantage of Fed-REACT over existing supervised learning baselines. Future work may explore the fully-decentralized setting where the clients need to learn models for time series data without the help of a coordinating server.

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

The appendix is structured as follows: Section A describes the steps of training task models while utilizing snapshot clustering; Section B presents details of Kalman smoothing and the EM algorithm; Section C provides the calculation of the forgetting factor $\alpha_t$ in the AFFECT algorithm; Section D contains detailed proofs of Lemmas and Theorem 1; Section E shows the experimental results on time-smoothed gradient descent.

## A    TASK MODEL TRAINING ASSISTED BY SNAPSHOT CLUSTERING

Snapshot clustering groups clients based on the current weights of the task model / output layer, and then averages those weights to arrive at a cluster-specific task model. This procedure is formalized as Algorithm 3 below. Note that snapshshot clustering may provide satisfactory performance when clients have exceedingly large number of labeled samples in training batches so that the models do not experience training variations.

---

**Algorithm 3** Training of the task model assisted by snapshot clustering

---

1: **Initialize:** Global encoder parameters $\theta_T$ obtained after $T$ rounds of federated representation learning presented in Alg. 1
2: **for** client $k = 1, 2, .., K$ **do**
3:      Client $k$ trains the task model on the labeled local data.
4:      Client $k$ uploads the parameters $\theta_{task}^k$ of the task model to the server
5: **end for**
6: Server clusters clients based on the weights of the task model $\{\theta_{task}^k\}_{k=1}^K$ and employs Agglomerative Hierarchical Clustering.
7: **for** cluster $c = 1, 2, .., C$ **do**
8:      Server aggregates the task model within cluster. Let $\mathcal{S}_t^c$ denote the set of clients in cluster $c$. Then

$$\theta_{\mathbf{task}}{}^c = \sum_{k \in \mathcal{S}_t^c} \frac{m_k}{M_c} \theta_{task}^k$$

where $m_k$ is the number of labeled samples on client $k$ and $M_c = \sum_{k \in \mathcal{S}_t^c} m_k$
9:      Server transmits $\theta_{\mathbf{task}}^{\mathbf{c}}$ to all clients $k \in \mathcal{S}_t^c$
10: **end for**

---

## B    KALMAN SMOOTHING AND THE EM ALGORITHM

### B.1    KALMAN SMOOTHER

Consider the following state space model relating states $x_t \in R^n$ and measurements $y_t \in R^m$:
$$x_t = F x_{t-1} + q_t, \tag{7}$$
$$y_t = H x_t + r_t. \tag{8}$$
The state equation matrix $F \in R^{n \times n}$ and the process noise $q_t \in \mathcal{N}(0, Q)$ drive the evolution of the hidden state across time whereas the measurement matrix $H \in R^{m \times n}$ and the measurement noise $r_t \in \mathcal{N}(0, R)$ drive the observability of the hidden state. In our experiments, we assume that the system parameters $F, H, Q, R$ remain constant over time. If the initial state $x_0$ is Gaussian, i.e., $x_0 \in \mathcal{N}(\mu_0, \Sigma_0)$, it can be shown that the minimum mean square error (MMSE) estimate of $x_t$ given measurements $y_1, ..., y_t$, denoted by $\hat{x}_t^+$ and equal to $E[x_t | y_1, y_2, ..., y_t]$, can be found as a linear combination of the measurements. In particular, the MMSE estimate $\hat{x}_t^+$ can be found via recursive expressions of the Kalman filter given below.

**Predict Step**:
$$\hat{x}_t^- = F \hat{x}_{t-1}^+ \tag{9}$$
$$\hat{P}_t^- = F \hat{P}_{t-1}^+ F^T + Q \tag{10}$$

**Update Step**:

$$K_t = \hat{P}_t^- H^T (H \hat{P}_t^- H^T + R)^{-1} \tag{11}$$

$$\hat{x}_t^+ = \hat{x}_t^- + K_t(y_t - H\hat{x}_t^-) \tag{12}$$

$$\hat{P}_t^+ = (I - K_t H)\hat{P}_t^-, \tag{13}$$

where $\hat{P}_t^- = E[||x_t - E[x_t|y_1, y_2, ..., y_{t-1}]||^2]$ and $\hat{P}_t^+ = E[||x_t - E[x_t|y_1, y_2, ..., y_t]||^2]$.

Once we have iterated through all the $N$ measurements available, we can perform Kalman Smoothing through a backward pass using the Rauch-Tung-Streiber (RTS) algorithm. Let $\hat{x}_{t|N}^+$ denote the smoothed estimate of $x_t$ given the measurements $y_1, ..., y_N$, and let $\hat{P}_{t|N}^+$ denote the corresponding error covariance. The backward pass is initialized with $\hat{x}_{N|N}^+ = \hat{x}_N^+$ and $\hat{P}_{N|N}^+ = \hat{P}_N^+$. Then

$$G_{t-1} = \hat{P}_{t-1}^+ F (\hat{P}_t^-)^{-1} \tag{14}$$

$$\hat{x}_{t-1|N}^+ = \hat{x}_{t-1}^+ + G_{t-1}(\hat{x}_{t|N}^+ - \hat{x}_t^-) \tag{15}$$

$$\hat{P}_{t-1|N}^+ = \hat{P}_{t-1}^+ + G_{t-1}(\hat{P}_{t|N}^+ - \hat{P}_t^-)G_{t-1}^T \tag{16}$$

As seen in Appendix B.2, the Expectation-Maximization algorithm requires calculation of *"lag one smoothed covariance"* defined as $\hat{P}_{t,t-1|N}^+ = E[(x_t - \hat{x}_{t|N}^+)(x_{t-1} - \hat{x}_{t-1|N}^+)^T|y_1, ..., y_N]$. The recursive equation for lag one smoothed covariance can be calculated as

$$\hat{P}_{N,N-1|N}^+ = (I - K_N H)F\hat{P}_{N-1}^+ \tag{17}$$

$$\hat{P}_{t,t-1|N}^+ = \hat{P}_{t-1}^+ G_{t-1}. \tag{18}$$

## B.2 EM ALGORITHM FOR KALMAN SMOOTHING

The authors in Shumway & Stoffer (1982) explore the estimation of $\Theta = \{F, Q, R\}$ in support of the state estimation using the Expectation-Maximization algorithm. Under the assumption of Gaussianity, the conditional expectation of the likelihood

$$E_{X|Y;\hat{\Theta}_r}[\log(P(y_1, ..., y_N, x_1, ..., x_N; \Theta))]$$

can be expressed as a function of $\{\hat{x}_{t|N}^{+(r)}\}_{t=1}^N$ which are conditioned not only on $y_1, ..., y_N$ but also on $\hat{\Theta}_r$ (the estimates of $F$, $R$, and $Q$ in the $r^{th}$ iteration of the EM algorithm). Setting the derivative of the resulting expression with respect to $\{F, Q, R\}$ to zero yields

$$F^{(r+1)} = BA^{-1} \tag{19}$$

$$Q^{(r+1)} = \frac{1}{N}(C - BA^{-1}B^T) \tag{20}$$

$$R^{(r+1)} = \frac{1}{N}\sum_{t=1}^N((y_t - H\hat{x}_{t|N}^{+(r)})(y_t - H\hat{x}_{t|N}^{+(r)})^T + H\hat{P}_{t|N}^{+(r)}H^T) \tag{21}$$

where

$$A = \sum_{t=1}^N \hat{P}_{t-1|N}^{+(r)} + \hat{x}_{t-1|N}^{+(r)}\hat{x}_{t-1|N}^{+(r)\,T} \tag{22}$$

$$B = \sum_{t=1}^N \hat{P}_{t,t-1|N}^{+(r)} + \hat{x}_{t|N}^{+(r)}\hat{x}_{t-1|N}^{+(r)\,T} \tag{23}$$

$$C = \sum_{t=1}^N \hat{P}_{t|N}^{+(r)} + \hat{x}_{t|N}^{+(r)}\hat{x}_{t|N}^{+(r)\,T}. \tag{24}$$

The EM algorithm then alternates between the estimates of the parameters $\Theta$ and the (smoothed) state estimates.

---

**Algorithm 4** Estimating $\alpha_t$ iteratively

---

1: **for** iteration $iter = 1, 2, .., MaxIterations$ **do**
2:     Estimate $\mathcal{S}_t^c$ given $\hat{\psi}_{i,j,t-1}$, $\hat{\alpha}_t$ which yield $[\hat{\psi}_t]_{i,j}$. In our work, this is done via Agglomerative Hierarchical Clustering.
3:     Compute $\hat{\mathbb{E}}[[W_t]_{i,j}]$ and $\hat{Var}([W_t]_{i,j})$ based on $\mathcal{S}_t^c$ as described above
4:     Estimate $\hat{\alpha}_t$ using equation (27).
5: **end for**

---

## C   CALCULATION OF THE FORGETTING FACTOR $\alpha_t$

For completeness, we here summarize the derivation of the adaptive forgetting factor presented in (Xu et al., 2014). Let $K$ denote the total number of clients, and let $L(\alpha)$ be the Frobenius norm of the difference between the estimated and the true similarity matrix, i.e.,

$$L(\alpha) = \|\psi_t - \alpha_t \hat{\psi}_{t-1} - (1 - \alpha_t) W_t\|_F^2 \tag{25}$$

Then the risk function $R(\alpha) = \mathbb{E}[L(\alpha)]$ can be shown to take the form

$$R(\alpha) = \sum_{i=1}^{K} \sum_{j=1}^{K} \{(1 - \alpha)^2 Var([W_t]_{i,j}) + \alpha^2 ([\hat{\psi}_t]_{i,j} - [\psi_{t-1}]_{i,j})^2\}, \tag{26}$$

where $[W_t]_{i,j}$, $[\hat{\psi}_t]_{i,j}$ and $[\psi_t]_{i,j}$ denote the entries at index $(i,j)$ of matrices $W_t$, $\hat{\psi}_t$ and $\psi_t$, respectively. To obtain this expression, it is assumed that $\mathbb{E}[[W_t]_{i,j}] = [\psi_t]_{i,j}$ and $Var([\psi_t]_{i,j}) = 0$. Taking the first derivative of $R(\alpha)$ w.r.t to $\alpha$ and setting it to zero yields

$$\hat{\alpha}_t = \frac{\sum_{i=1}^{K} \sum_{j=1}^{K} Var([W_t]_{i,j})}{\sum_{i=1}^{K} \sum_{j=1}^{K} ([\hat{\psi}_t]_{i,j} - [\psi_t]_{i,j})^2 + Var([W_t]_{i,j})}. \tag{27}$$

Note that the calculation in (27) requires $\mathbb{E}[[W_t]_{i,j}]$ and $Var([W_t]_{i,j})$, which in turn requires knowledge of the clustering solution $\mathcal{S}_t^c$, which depends on $\alpha_t$. Xu et al. (2014) proposed to estimate $\mathbb{E}[[W_t]_{i,j}]$, $Var([W_t]_{i,j})$ and $\alpha_t$ iteratively. Suppose client $l$ is assigned to cluster $c$; then for $j \neq l$,

$$\hat{\mathbb{E}}[[W_t]_{i,j}] = \sum_{i=l} \sum_{j \in c, j \neq l} \frac{1}{|c||c-1|} [W_t]_{i,j} \tag{28}$$

and

$$\hat{\mathbb{E}}[[W_t]_{i,j}] = \sum_{i=1}^{C} \frac{1}{C} W_{i,i}. \tag{29}$$

For $k$ and $l$ in distinct clusters $c$ and $d$, respectively, it holds that

$$\hat{\mathbb{E}}[[W_t]_{k,l}] = \sum_{i \in c} \sum_{j \in d} \frac{1}{|c||d|} [W_t]_{i,j}. \tag{30}$$

Estimates of the variances can be computed in a similar manner and are thus omitted for the sake of brevity. The resulting procedure is formalized as Algorithm 4. In our simulations, we set the number of iterations to 5.

## D  PROOF OF LEMMAS AND THEOREM

Using $\beta$-smoothness assumption of $f_{t,k}$ functions, it can be shown that $S_t$ is $\beta$-smooth. Then we have

$$S_{t,w,\gamma}(\theta_{t+1}) - S_{t,w,\gamma}(\theta_t) = \frac{1}{K}\sum_{k=1}^{K} S_{t,w,\gamma,k}(\theta_{t+1}) - S_{t,w,\gamma,k}(\theta_t)$$

$$\leq \frac{1}{K}\sum_{k=1}^{K}\langle \nabla S_{t,w,\gamma,k}(\theta_t), \theta_{t+1} - \theta_t\rangle + \frac{\beta}{2}\|\theta_{t+1} - \theta_t\|^2$$

$$= \langle \nabla S_{t,w,\gamma}(\theta_t), \theta_{t+1} - \theta_t\rangle + \frac{\beta}{2}\|\theta_{t+1} - \theta_t\|^2$$

$$= -\frac{\eta}{2}\langle \nabla S_{t,w,\gamma}(\theta_t), \tilde{\nabla} S_{t,w,\gamma}(\theta_t) + \epsilon_{proj}\rangle - \frac{\eta}{2}\langle \nabla S_{t,w,\gamma}(\theta_t), \tilde{\nabla} S_{t,w,\gamma}(\theta_t) + \epsilon_{proj} - \nabla S_{t,w,\gamma}(\theta_t)\rangle$$

$$- \frac{\eta}{2}\|\nabla S_{t,w,\gamma}(\theta_t)\|^2 + \frac{\eta^2\beta}{4}\|\tilde{\nabla} S_{t,w,\gamma}(\theta_t) + \epsilon_{proj} - \nabla S_{t,w,\gamma}(\theta_t) + \nabla S_{t,w,\gamma}(\theta_t)\|^2$$

$$+ \frac{\eta^2\beta}{4}\|\tilde{\nabla} S_{t,w,\gamma}(\theta_t) + \epsilon_{proj}\|^2$$

where $\epsilon_{proj}$ represents the projection error.

Therefore,
$$S_{t,w,\gamma}(\theta_{t+1}) - S_{t,w,\gamma}(\theta_t)$$

$$\leq -(\frac{\eta}{2} - \frac{\eta^2\beta}{4})\|\nabla S_{t,w,\gamma}(\theta_t)\|^2 - (\frac{\eta}{2} - \frac{\eta^2\beta}{4})\langle \nabla S_{t,w,\gamma}(\theta_t), \tilde{\nabla} S_{t,w,\gamma}(\theta_t) + \epsilon_{proj} - \nabla S_{t,w,\gamma}(\theta_t)\rangle$$

$$+ \frac{\eta^2\beta}{4}\|\tilde{\nabla} S_{t,w,\gamma}(\theta_t) + \epsilon_{proj} - \nabla S_{t,w,\gamma}(\theta_t)\|^2$$

$$\leq -(\frac{\eta}{2} - \frac{\eta^2\beta}{4})\|\nabla S_{t,w,\gamma}(\theta_t)\|^2 - (\frac{\eta}{2} - \frac{\eta^2\beta}{4})\langle \nabla S_{t,w,\gamma}(\theta_t), \tilde{\nabla} S_{t,w,\gamma}(\theta_t) - \nabla S_{t,w,\gamma}(\theta_t)\rangle$$

$$- (\frac{\eta}{2} - \frac{\eta^2\beta}{4})\langle \nabla S_{t,w,\gamma}(\theta_t), \epsilon_{proj}\rangle + \frac{\eta^2\beta}{2}\|\tilde{\nabla} S_{t,w,\gamma}(\theta_t) - \nabla S_{t,w,\gamma}(\theta_t)\|^2 + \frac{\eta^2\beta\epsilon^2}{2}$$

$$\leq -\frac{1}{2}(\frac{\eta}{2} - \frac{\eta^2\beta}{4})\|\nabla S_{t,w,\gamma}(\theta_t)\|^2 - (\frac{\eta}{2} - \frac{\eta^2\beta}{4})\langle \nabla S_{t,w,\gamma}(\theta_t), \tilde{\nabla} S_{t,w,\gamma}(\theta_t) - \nabla S_{t,w,\gamma}(\theta_t)\rangle$$

$$+ \frac{1}{2}(\frac{\eta}{2} - \frac{\eta^2\beta}{4})\epsilon^2 + \frac{\eta^2\beta}{2}\|\tilde{\nabla} S_{t,w,\gamma}(\theta_t) - \nabla S_{t,w,\gamma}(\theta_t)\|^2 + \frac{\eta^2\beta\epsilon^2}{2}.$$

By applying the conditional expectation $\mathbb{E}[\cdot|\theta_t]$ to both sides of the inequality, we obtain

$$(\frac{\eta}{4} - \frac{\eta^2\beta}{8})\|\nabla S_{t,w,\gamma}(\theta_t)\|^2$$

$$\leq \mathbb{E}[S_{t,w,\gamma}(\theta_t) - S_{t,w,\gamma}(\theta_{t+1})] + \eta^2\frac{\beta}{2}\frac{\sigma^2(1-\gamma^{2w})}{W^2(1-\gamma^2)} + (\frac{\eta}{4} - \frac{\eta^2\beta}{8} + \frac{\eta^2\beta}{2})\epsilon^2$$

$$= S_{t,w,\gamma}(\theta_t) - S_{t+1,w,\gamma}(\theta_{t+1}) + S_{t+1,w,\gamma}(\theta_{t+1}) - S_{t,w,\gamma}(\theta_{t+1}) + \eta^2\frac{\beta}{4}\frac{\sigma^2(1-\gamma^{2w})}{W^2(1-\gamma^2)}$$

$$+ (\frac{\eta}{4} - \frac{\eta^2\beta}{8} + \frac{\eta^2\beta}{2})\epsilon^2.$$

Rearranging the left and right side terms gives the inequality in Lemma 1.

Next, we derive the upper bounds for $S_{t+1,w,\gamma}(\theta_{t+1}) - S_{t,w,\gamma}(\theta_{t+1})$ and $S_{t,w,\gamma}(\theta_t) - S_{t+1,w,\gamma}(\theta_{t+1})$. Recall that each loss function $f_t$ is upper bounded by $M$, i.e., $|f_t(x)| \leq M$. Then

$$S_{t+1,w,\gamma}(\theta_{t+1}) - S_{t,w,\gamma}(\theta_{t+1}) = \frac{1}{W}\sum_{j=0}^{w-1}\gamma_j(f_{t+1-j}(\theta_{t+1-j}) - f_{t-j}(\theta_{t+1-j}))$$

$$= \frac{1}{W}[f_{t+1}(\theta_{t+1}) - f_t(\theta_{t+1}) + \gamma f_t(\theta_t) - \gamma f_{t-1}(\theta_t) + \cdots$$

$$+ \gamma^{w-1}f_{t-w+2}(\theta_{t-w+2}) - \gamma^{w-1}f_{t-w+1}(\theta_{t-w+2})]$$

$$\leq \frac{M(1+\gamma^{w-1})}{W} + \frac{M(1-\gamma^{w-1})(1+\gamma)}{W(1-\gamma)}$$

$$S_{t,w,\gamma}(\theta_t) - S_{t+1,w,\gamma}(\theta_{t+1}) = \frac{1}{W}\sum_{j=0}^{w-1}\gamma^j(f_{t-j}(\theta_{t-j}) - f_{t+1-j}(\theta_{t+1-j}))$$

$$\leq \frac{2M(1-\gamma^w)}{W(1-\gamma)}$$

This completes the proof of Lemma 2 and 3.

Using the inequalities above, we derive an upper bound on $\|\nabla S_{t,w,\gamma}(\theta_t)\|^2$ as

$$\|\nabla S_{t,w,\gamma}(\theta_t)\|^2$$

$$\leq \frac{\frac{2M(1-\gamma^w)}{W(1-\gamma)} + \frac{M(1+\gamma^{w-1})}{W} + \frac{M(1-\gamma^{w-1})(1+\gamma)}{W(1-\gamma)} + \eta^2\frac{\beta}{4}\frac{\sigma^2(1-\gamma^{2w})}{W^2(1-\gamma^2)} + (\frac{\eta}{4} - \frac{\eta^2\beta}{8} + \frac{\eta^2\beta}{2})\epsilon^2}{(\frac{\eta}{4} - \frac{\eta^2\beta}{8})}.$$

Substituting $\eta = \frac{1}{\beta}$ yields

$$\|\nabla S_{t,w,\gamma}(\theta_t)\|^2$$

$$\leq \frac{8\beta M}{W}(\frac{2(1-\gamma^w)}{1-\gamma} + (1+\gamma^{w-1}) + \frac{(1-\gamma^{w-1})(1+\gamma)}{1-\gamma}) + \frac{2\sigma^2(1-\gamma^{2w})}{W^2(1-\gamma^2)} + \frac{5}{8}\epsilon^2$$

$$\leq \frac{8\beta M}{W}(\frac{2(1-\gamma^w)}{1-\gamma} + (1+\gamma^{w-1}) + \frac{(1-\gamma^w)(1+\gamma)}{1-\gamma}) + \frac{2\sigma^2(1-\gamma^{2w})}{W^2(1-\gamma^2)} + \frac{5}{8}\epsilon^2$$

$$= \frac{8\beta M}{W}(\frac{(1-\gamma^w)(3+\gamma)}{1-\gamma} + (1+\gamma^{w-1})) + \frac{2\sigma^2(1-\gamma^{2w})}{W^2(1-\gamma^2)} + \frac{5}{8}\epsilon^2$$

$$\leq \frac{8\beta M}{W}(4\frac{1-\gamma^w}{1-\gamma} + \frac{1+\gamma^{w-1}}{1-\gamma}) + \frac{2\sigma^2(1-\gamma^{2w})}{W^2(1-\gamma^2)} + \frac{5}{8}\epsilon^2$$

$$\leq \frac{32\beta M}{W}(\frac{2-\gamma^w+\gamma^{w-1}}{1-\gamma}) + \frac{2\sigma^2(1-\gamma^{2w})}{W^2(1-\gamma^2)} + \frac{5}{8}\epsilon^2.$$

When $\gamma \to 1^-$,

$$\lim_{\gamma\to 1^-}\|\nabla S_{t,w,\gamma}(\theta_t)\|^2 \leq \frac{1}{W}(64\beta M + 2\sigma^2) + \frac{5}{8}\epsilon^2.$$

Telescoping $t$ from 1 to $T$, we obtain

$$\lim_{\gamma\to 1^-}\sum_{t=1}^{T}\|\nabla S_{t,w,\gamma}(\theta_t)\|^2 \leq \frac{T}{W}(64\beta M + 2\sigma^2) + \frac{5}{8}\epsilon^2 T$$

and

$$\lim_{\gamma\to 1^-}\frac{1}{T}\sum_{t=1}^{T}\|\nabla S_{t,w,\gamma}(\theta_t)\|^2 \leq \frac{1}{W}(64\beta M + 2\sigma^2) + \frac{5}{8}\epsilon^2$$

This concludes the proof of Theorem 1.

## E    EXPERIMENTAL RESULTS ON TIME-SMOOTHED GRADIENT DESCENT

The time-smoothed gradient descent algorithm DTSSGD, proposed by Aydore et al. (2019), presents a regret framework for non-convex models that deals with the concept drift associated with a dynamic environment. We compare our results with those obtained by training the encoder using DTSSGD. The experiments are conducted on the RTD dataset with ten clients partitioned into 3 clusters created

using Dirichlet sampling ($\beta = 0.1$). As before, the encoder was trained for 10 rounds but with the optimizer set to the one proposed in (Aydore et al., 2019). Training of the output layer consists of a single round involving all the labeled samples available at a client. We vary the parameter $\gamma$ (used to control forgetting) and the smoothing window size $w$. The results are presented in Table 5.

| $\gamma$ | $w = 1$ | $w = 3$ | $w = 5$ | $w = 7$ |
|------|-------|-------|-------|-------|
| 0.7 | 0.988 | 0.984 | 0.988 | 0.986 |
| 0.8 | 0.988 | 0.990 | 0.982 | 0.985 |
| 0.9 | 0.988 | 0.980 | 0.990 | 0.990 |

Table 5: Results for Fed-REACT using the optimizer from (Aydore et al., 2019).

The results suggest that increasing $w$ does not lead to significant performance gain; therefore, in our experiments we set $w = 1$.

