# OpenReview forum: "Fed-REACT: Federated Representation Learning for Heterogeneous Time Series Data"
_ICLR.cc/2025/Conference — Submitted to ICLR 2025_

### Official Review · Reviewer_K5Dg · 2024-11-03

**Soundness:** 2
**Presentation:** 3
**Contribution:** 3
**Rating:** 6
**Confidence:** 3

**Summary:**

The paper introduces a novel federated learning framework designed to address the challenges of heterogeneous time series data across multiple clients. The proposed method, Fed-REACT, consists of a two-stage learning process: the first stage focuses on self-supervised learning to extract meaningful features from local time series data, while the second stage employs evolutionary clustering to adaptively group clients and train task models that reflect data heterogeneity. The authors present an adaptive evolutionary clustering approach that enhances stability by incorporating both current and historical model weights. Through theoretical analysis and experimental validation on given datasets, the paper demonstrates that Fed-REACT improves model performance compared to existing methods.

**Strengths:**

1. The paper is well-structured and well-written, making it easy to understand.
2. The paper provides a novel approach that combines representation learning and evolutionary clustering.
3. The paper also gives a theoretical analysis of the proposed method, including convergence properties and a global regret function.
4. The authors include a thorough experimental analysis on the given datasets demonstrating the effectiveness of Fed-REACT.

**Weaknesses:**

1. While the paper discusses the necessity of federated learning in non-IID settings, it does not include detailed experimental results. Only results on one dataset do not help generalize the performance of the method over non-iid.
2. A detailed privacy analysis of the proposed method is missing which is very important in a federated setting.
3. A more thorough literature review would be nice, including state-of-the-art methods in time series federated learning

**Questions:**

1. How does the proposed method perform in non-IID data distributions among various clients (include more datasets)?
2. Can you provide a detailed privacy analysis of the proposed methods, and how do you plan to address this important aspect in the context of federated learning?

---

> ### Author Response · Authors · 2024-12-01
> **Reply to reviewer K5Dg**
>
> We appreciate the reviewer’s valuable comment and kindly refer to our responses provided below.
>
> Weakness 1: In the experimental section, we included two datasets, both designed to capture non-IID settings. For the first dataset, the RTD dataset, we assigned temporal sequences with different labels to clients, ensuring that the data distributions were non-IID and evolved over time. To evaluate the performance of Fed-REACT in this non-IID scenario, we tested it under varying client numbers (10, 50, and 100) and different levels of heterogeneity, controlled by the Dirichlet parameter β.
> For the second dataset, the SUMO dataset, the data was collected from different vehicles, inherently introducing non-IID characteristics due to variations in vehicle-specific data. We demonstrated the efficacy of the Fed-REACT algorithm in this naturally non-IID scenario as well. These experiments validate the robustness of Fed-REACT across diverse non-IID conditions and in the future we will include additional datasets to further validate its generalizability.
>
> Weakness 2: Thank you for raising the important concern regarding privacy. In our proposed Fed-REACT framework, only model parameters are transmitted across the network, with no raw data or representations of raw data being shared. This design minimizes the risk of exposing raw data or local data distributions. Additionally, when updating local models, we utilize stochastic gradients, which help mitigate the risk of data leakage through gradients, thereby enhancing privacy preservation in the federated learning setting. More details and further enhancements are included in responses to Question 2.
>
> Weakness 3: We thank the reviewer for the valuable feedback. In response, we have updated the manuscript to include a more thorough literature review, particularly incorporating recent advancements in time series.
>
> Question 1: The proposed method is designed to handle non-IID data distributions effectively, and we evaluated its performance on two datasets, both capturing non-IID scenarios.
>
> 1.	RTD Dataset: To simulate non-IID conditions, temporal sequences with different labels were assigned to clients, resulting in non-IID and evolving data distributions. We further evaluated the method's performance under varying client numbers (10, 50, and 100) and different levels of heterogeneity, controlled by the Dirichlet parameter (β). The results demonstrate that Fed-REACT performs robustly across these non-IID conditions.
>
> 2.	SUMO Dataset: This dataset inherently exhibits non-IID characteristics, as the data is collected from different vehicles, each representing unique local distributions. Fed-REACT effectively handled this naturally non-IID scenario, showcasing its adaptability and efficacy.
>
> While these two datasets highlight the method's effectiveness in non-IID settings, we acknowledge the value of incorporating additional datasets to further validate its generalizability and will prioritize this in future work.
>
> Question 2: Thank again for highlighting the importance of privacy in federated learning. Below is a detailed analysis of the privacy considerations in our proposed Fed-REACT framework and our plans to enhance privacy further:
>
> Privacy Analysis:
>
> 1.	Model Parameter Sharing: In Fed-REACT, only model parameters are transmitted between the server and clients. No raw data or representations of raw data are shared, which significantly reduces the risk of exposing sensitive information about the clients’ local data.
>
> 2.	No Representation Sharing: Unlike some methods that share intermediate representations, Fed-REACT avoids transmitting representations of local data, further enhancing privacy by reducing the attack surface.
>
> 3.	Gradient-Based Updates: Local model updates are computed using stochastic gradients. This inherently adds noise to the updates, making it more challenging for an adversary to infer sensitive information from the transmitted gradients. This approach mitigates risks such as gradient inversion attacks.
>
> Further enhancement:
>
> To further enhance privacy guarantees, we can integrate additional strategies into the proposed Fed-REACT algorithm, including: (1) applying differential privacy by adding noise to the model updates, and (2) implementing adversarial training mechanisms to ensure that the learned representations are robust against adversarial attacks.
>
> Thanks again for all thoughtful feedback and we hope our revisions and explanations address all your concerns effectively.

---

### Official Review · Reviewer_61No · 2024-11-03

**Soundness:** 2
**Presentation:** 2
**Contribution:** 2
**Rating:** 3
**Confidence:** 2

**Summary:**

This paper investigates the federated representation learning problem for time series data. A new method based on evolutionary clustering is proposed to handle the problem. Theoretical analysis is given to discuss the properties of the proposed method. Experiments are conducted to show the effectiveness of the proposed method.

**Strengths:**

[1] Sufficient theoretical analysis is provided, which provides a solid guarantee of the usefulness of the proposed method.

[2] The idea of balancing the contribution of structure and feature information makes sense, especially when the missing situation is unknown or uncertain.

**Weaknesses:**

[1] Experiments are carried out on only two datasets. More datasets should be considered to lead to comprehensive performance results.

[2] There should be more discussion on the unique challenges in time series representation learning and how existing federated learning/federated representation learning methods fail to perform well.

[3] Usually, representation learning aims to learn high-quality representations that can be utilized in various downstream tasks. However, this paper only discusses the optimization for the SVM classifier, which is not aligned with the goal of representation learning.

[4] More up-to-date baselines should be compared with the proposed method in the experimental part.

**Questions:**

N/A

---

> ### Author Response · Authors · 2024-11-26
> **Reply to reviewer 61No**
>
> We thank the reviewer for the valuable comments and please see below our responses to all comments:
>
> Weakness [1]: Regarding the number of datasets used in our experiments, we appreciate your emphasis on comprehensive performance evaluation. Due to time and resource constraints, we were unable to include additional datasets in current manuscript. However, we carefully selected the two datasets to ensure they represent diverse and challenging scenarios relevant to the studied problem. With the two datasets, we validated the efficacy of the proposed algorithm under different heterogeneity levels and network sizes. We acknowledge that including more datasets would further validate the generalizability of our approach, and we aim to address this in future work.
>
>
> Weakness [2]:
> 2.1. This work formally examines the federated learning problem where clients possess heterogeneous time series data. This is a complex challenge due to two distinct sources of data heterogeneity in such FL systems: inter-client distribution diversity, which arises from variations in data distributions across clients, and intra-client data heterogeneity, characterized by the potential non-stationarity of data observed locally by each client. While inter-client heterogeneity is addressed through representation learning on time series data, this approach alone does not resolve intra-client distribution variability. The second phase of our proposed algorithm is designed to tackle this specific challenge.
>
> 2.2. In comparison to existing methods, most clustered FL approaches (e.g., IFCA, FLSC) fail to account for the evolving nature of local data distributions over time, leading to issues when distribution shifts cause clients to move between clusters. Conversely, most time series FL methods (e.g., EFDLS, FedTADBench) overlook either the possibility that clients may have heterogeneous distributions or the possibility that clients with fewer local data samples can learn from clusters, limiting their applicability in such settings.
>
>
> Weakness [3]: Our proposed method is applicable to various downstream tasks. Specifically, in Phase 2 of the proposed algorithm, Fed-REACT, the task model update is provided in a manner that supports different types of task models, not limited to SVM classifiers. We have revised the manuscript to ensure that the statements are consistent with the method's applicability to diverse downstream tasks, not limited to classification tasks. In Section 4.2, for example, we address a regression task using linear output layers as the downstream task models.
>
>
> Weakness [4]: Regarding more up-to-date baselines, we will prioritize including more baselines in the broader domain in our future work. In the current version, we carefully selected existing baselines that represent state-of-the-art methods closely aligned with our specific problem, especially as we are addressing a novel issue. Moving forward, we will incorporate a broader range of baselines in related domains as part of our future work. If the reviewer has any recommendations, we'd love to add them as baseline algorithms in our future work.
>
>
> Thanks for all the valuable comments and we hope our responses can address your concerns.
>
>
> References:
> IFCA - An efficient framework for clustered federated learning
>
> FLSC - Federated learning with soft clustering
>
> EFDLS - An Efficient Federated Distillation Learning System for Multitask Time Series Classification
>
> FedTADBench - FedTADBench: Federated Time-series Anomaly Detection Benchmark

---

### Official Review · Reviewer_bjq8 · 2024-11-04

**Soundness:** 2
**Presentation:** 1
**Contribution:** 1
**Rating:** 3
**Confidence:** 3

**Summary:**

This study presents Fed-REACT, a federated learning method tailored for heterogeneous time series data, integrating representation learning and evolutionary clustering. The method operates in two stages: first, clients employ self-supervised learning to extract features from their local data; second, the server uses evolutionary clustering with an adaptive forgetting factor to cluster clients based on their features. Clients from the same cluster collaborate in order to learn the parameters of a downstream task.

The paper provides a theoretical analysis for self-supervised feature extraction, when a rank one approximation of  the covariance matrix is employed.

The paper provides numerical simulations demonstrating the effectiveness of the proposed approach, in particular in settings with a small number of clients.

**Strengths:**

* The paper provides a reasonable number of numerical simulations in order to evaluate the performance of the proposed approach.
* The paper does not have any major errors, and most of the claims are correct.

**Weaknesses:**

* The paper is overall hard to follow due to the confusion notation. For example, (1) introduces the contrastive loss that would be used in the first phase of feature extraction. However, the theoretical analysis does not make any reference to (1); instead it introduces an other objective function, $f_{\text{SSL}}$. An other example of inconsistent notation, is the introduction of $\phi_t$, $W_t$, and $N_t$, which are not explicitly connected to the parameter  of the problem $\theta$ and $\theta_{\text{task}}$.
* I do not really see how Section 3 fits in the paper and how it serves its goal. This section proves the convergence of federated projected gradient descent under a restrictive set of assumptions to a neighborhood of a stationary point.
* The numerical simulations show the advantage of the proposed method when the number of clients is small, but the advantage of this method is small when the number of clients is relatively large (>50).
* The paper is principally proposing to learn a feature extractor in a federated self-supervised fashion. Afterwards, evolutionary clustering is used to learn one model by cluster. Both these techniques are already known, therefore the novelty of the paper is limited.
* The proposed approach is claimed to be tailored for time series, but it could be used for any type of data.

**Questions:**

Can you please explain how is the proposed approach specific to time series?

---

> ### Author Response · Authors · 2024-11-13
> **Reply to Reviewer bjq8**
>
> We thank the reviewer for the valuable feedback. Please see our responses below:
>
> Weakness 1.1: Contrastive Loss Definition — The contrastive loss defined in (1) is used to develop Phase 1 of the Fed-REACT algorithm and is applied in our experiments. In the theoretical analysis (Section 3), we use a simplified version of the contrastive loss (1), focusing on positive pairs to make the analysis more tractable, as done in previous representation learning literature (e.g., $\textit{Does learning from decentralized non-iid unlabeled data benefit from self-supervision?}$). This purpose of the analysis is to provide theoretical insights into time-series representation learning in a federated learning setting.
>
> Weakness 1.2: Notation Consistency — Notations vary throughout the manuscript for clarity: $\theta$ represents the feature extractor model parameters, while $\theta_{task}$ represents task model parameters. In Phase 2, $W_t$, $\psi_t$, and $N_t$ are used for evolutionary clustering in task model updates. Here, $W_t$ denotes the similarity matrix of task model parameters, with each element representing the cosine similarity of parameters from two clients, as specified in lines 177–178.
>
> Weakness 2: Purpose of Section 3 — This work addresses federated learning on heterogeneous time-series data, leveraging self-supervised learning to capture meaningful temporal features. Section 3 aims to offer theoretical insights into federated self-supervised learning on time-series data. Specifically, we analyze a time-varying local objective function, $f_{t, i}$, and adopt time-smoothed gradient descent for time-series settings. We show that the regret converges to a small error, supporting the feasibility of federated self-supervised learning on time-series data.
>
> Weakness 3: Performance with Large Client Numbers — For scenarios with 100 clients, in Figure 3, the Fed-REACT algorithm achieves the highest Rand Score = 1 within just three communication rounds, converging to the true clustering membership. In contrast, baseline methods fail to identify the true clusters, with Rand Scores fluctuating with average below 0.8. Additionally, Table 3 shows that baseline methods (IFCA and FLSC) fall short of Fed-REACT across one or multiple distribution parameter values, $\beta$, demonstrating Fed-REACT’s efficacy in federated self-supervised learning on heterogeneous time-series data. The Phase 2 design in Fed-REACT further improves accuracy, achieving consistent results across all $\beta$ values.
>
> Weakness 4: Limited Novelty — To the best of our knowledge, this work is the first to address the novel problem of federated self-supervised learning on heterogeneous time-series data. For this problem, we specifically propose Fed-REACT that contains the federated self-supervised learning phase and the task model learning phase. The federated self-supervised learning framework has not been explored in heterogeneous time series data. The evolutionary clustering, has not been explored in the federated learning setting, either, since most work does not consider the clients’ models evolve over time. Phase 2’s adaptive forgetting factor further enhances evolutionary clustering, distinguishing Fed-REACT from simple combinations of existing methods.
>
> Weakness 5: Specification to Time-Series Data — Does $\textit{any type of data}$ refer to static data? Time-series data, with its temporal dimension, is a generalization of static data and requires specific consideration. Most FL algorithms focus on static data and do not extend naturally to time-series data. Fed-REACT is specifically designed for time-series data and can also handle static data as a special case by excluding the temporal dimension.
>
> Question 1: Specification to Time-Series Data — Fed-REACT is tailored for federated learning on heterogeneous time-series data, with four unique aspects:
>
> 1.	Phase 1 leverages contrastive learning, with positive/negative pairs selected within time-series trajectories.
>
> 2.	Phase 2 applies evolutionary clustering and adaptive forgetting, both critical for evolving local models in time-series contexts.
>
> 3.	Clients collect streaming time-series data for input into local models, resulting in a time-varying objective function.
>
> 4.	The optimization adopts time-smoothed gradient descent to address the time-varying objective function.
>
> Thank you for the thoughtful feedback. We hope these responses clarify our approach and contributions.

---

### Official Review · Reviewer_D1WC · 2024-11-04

**Soundness:** 3
**Presentation:** 2
**Contribution:** 3
**Rating:** 5
**Confidence:** 4

**Summary:**

The authors proposed a method Fed-REACT to tackle the time-series analysis in federated learning. They adopt self-supervised training in the first stage to learn useful representations. In the second stage, they use evolutionary clustering to cluster clients for local downstream tasks. Extensive experiments are conducted to evaluate the performance of the proposed mechanism.

**Strengths:**

1. Overall, the paper is well-written with a clear presentation.
2. Investigating time-series analysis in FL is a significant problem and direction.
3. Extensive experiments have been conducted to evaluate the performance of the proposed framework.

**Weaknesses:**

1. The explanation of motivation in the introduction is unclear. The reason for applying clustering and self-supervised learning to investigate time-series analysis in FL could be clarified.
2. The novelty of Weighted Averaging with Forgetting is limited. It is quite common to do clustering or aggregation by clients’ current and historical weights.
3. The compared baselines are too old. Some SOTA works about time-series FL or clustered FL could be considered.
4. As a heuristic algorithm, what is the efficiency and cost performance of evolutionary clustering?
5. The author could consider comparing with SOTA time series analysis models, such as Time-LLM, PatchTST, TimesNet, etc.
6. Are the experimental results repeated multiple times? Are there any standard deviation results shown?

**Questions:**

Please refer to weakness.

---

> ### Author Response · Authors · 2024-12-01
> **Reply to reviewer D1WC**
>
> We thank the reviewer for the valuable comments and please see below our responses:
>
> Weakness 1: Regarding the reason for clustering and self-supervised learning to investigate time-series analysis in FL, we have clarified this in the updated contribution section. This work is the first to formally investigate the problem of federated self-supervised learning on heterogeneous time-series data. There are two sources of data heterogeneity in such FL systems: Inter-client distribution diversity, arising from the differences in data distribution across clients, and intra-client data heterogeneity, i.e., potential non-stationarity of the data observed locally by each client. Our proposed method, Fed-REACT, consists of two learning phases: In the first phase, which essentially deals with inter-client data heterogeneity, the clients rely on self-supervised learning to collaboratively learn meaningful features, while in the second phase, addressing intra-data heterogeneity, temporally-evolving clusters of distributionally similar clients use the extracted features to train task (i.e., post-representation) models.
>
> Weakness 2: We appreciate the reviewer's comment regarding the novelty of the Weighted Averaging with Forgetting component. We would like to clarify how this component in our proposed Fed-REACT algorithm differs from prior work in the following ways:
>
> 1. Design for Cluster Center Updates: In the Fed-REACT algorithm, the Weighted Averaging with Forgetting mechanism is specifically designed for updating cluster center model parameters, rather than being applied directly to the clients’ model parameters, as is common in other methods.
>
> 2. Two-Step Application in Phase 2: The Weighted Averaging with Forgetting is employed in two distinct steps within Phase 2 of Fed-REACT:
> Step 1: It is used to compute an evolutionary similarity matrix to determine cluster memberships. This step leverages evolutionary clustering to dynamically group clients based on the similarity of their task model weights.
> Step 2: It is used to compute cluster center models. To address the challenge of weight variations (especially when training batches are small), we introduce an adaptive forgetting factor in this step. This factor enables the clustering process to account for both current and historical weights of task models, resulting in more accurate and stable clustering solutions.
>
> 3. Avoiding Client-Side Clustering Costs: In the context of clustered federated learning, many prior works (e.g., [1], [2]) determine cluster membership at the client side. This typically requires clients to download all cluster center models constantly, leading to increased communication and computational overhead. Our approach avoids these drawbacks by performing clustering centrally, reducing client-side resource requirements.
>
> Weakness 3: We sincerely thank the reviewer for highlighting the importance of including more recent baseline algorithms. As this study addresses a novel problem, there are currently no existing methods specifically designed for the problem under investigation. Therefore, our focus was on comparing our approach with methods that could be directly adapted to this context. In future work, we will consider incorporating a broader range of baselines, particularly in the areas of time series federated learning and clustered federated learning.
>
> Weakness 4: Thanks for the question and we compare the efficiency/cost from two aspects:
>
> 1. Our method employs evolutionary clustering at the central server, eliminating client-side clustering costs. In contrast, many prior works determine cluster memberships on the client side. This approach typically requires clients to download all cluster center models constantly, resulting in significant communication and computational overhead. By performing clustering centrally, our method avoids these issues and reduces resource requirements for clients.
>
> 2. Compared to snapshot clustering at the central server, evolutionary clustering offers improved accuracy by dynamically adapting to changes in client distributions. This dynamic adaptation leads to more stable and accurate model performance over time. Although this improvement may come at the cost of additional computational and communication overhead, in practice the computational cost can be lowered by reducing the density of similarity calculations.
>
> Weakness 5: We appreciate the reviewer’s suggestion regarding the inclusion of different time series baselines and will carefully consider and select those most suitable for the studied context in the future.
>
>
>
> [1]: An Efficient Framework for Clustered Federated Learning
>
> [2]: Dynamic Clustering in Federated Learning

---

> ### Author Response · Authors · 2024-12-01
> **Reply to reviewer D1WC**
>
> Weakness 6: Thank you for raising this important aspect. While the full experiments presented in the manuscript were conducted only once due to time and resource constraints, we did perform multiple trial runs beforehand to ensure that the performance was consistent and not subject to significant variability. These trial runs gave us confidence in the robustness of the results before proceeding with the final experiments. However, we recognize the importance of repeating the full experiments and reporting standard deviation to provide a comprehensive evaluation of the method's reliability. We will prioritize this in future work.
>
> We hope our responses address all your questions and we would love to discuss if you have further concerns.

---

### Meta-Review · Area_Chair_JKiF · 2024-12-19

**Metareview:**

The authors have made an attempt to address the reviewers' concerns. However, the rebuttal appears insufficient to convince the reviewers. This is mostly due to the (still) lack of empirical studies. The paper has also not been revised properly to address the reviewers' concern. As such, this paper does not merit an acceptance.

**Additional Comments On Reviewer Discussion:**

Overall, due to the original low ratings, there is little engagement between the authors and reviewers. The AC has asked the reviewers to have a final look into the authors' rebuttal but there is ultimately no change towards a more positive stance. One reviewer pointed out the remaining issues of this paper and the fact that the paper has not been properly revised to address the reviewers' concerns.

---

### Decision · Program_Chairs · 2025-01-22

Reject